# The Polychromatic Woodburytype—Colour Tracking in Translucent, Patterned Gelatin/Pigment Films

**DOI:** 10.3390/molecules25112468

**Published:** 2020-05-26

**Authors:** Damien Jon Leech, Walter Guy, Susanne Klein

**Affiliations:** Centre for Fine Print Research, University of the West of England, Bristol BS3 2JT, UK; walter.guy@uwe.ac.uk (W.G.); susanne.klein@uwe.ac.uk (S.K.)

**Keywords:** printing, colour reproduction, gelatin, woodburytype, Kubelka–Munk theory, photography, optical properties

## Abstract

The Woodburytype is a 19th century photomechanical technique capable of producing high-quality continuous-tone prints. It uses pigment dispersed in gelatin to produce a 2.5D print, in which the effect of varying tone is produced by a variation in the print height. We propose a method of constructing full colour prints in this manner, using a CMY colour model. This involves the layering of multiple translucent pigmented gelatin films and tracking how the perceived colour of these stacks changes with varying height. A set of CMY inks is constructed, taking into account the optical properties of both the pigment and gelatin, and a method of translating images into these prints is detailed.

## 1. Introduction

There is a difference between 3D printing coloured parts and 3D printing of full colour parts. Printing coloured parts means the colour of the printing material is used to construct a coloured object, with a limited colour palette. A full colour part instead mimics the colour of the existing object that it is replicating. There are only few 3D printers that claim to be able to print full colour. Alongside this, only a handful of articles attempt to explore printing photorealistic colour and the colour gamut of 3D printing apparatus [1,2,3,4]. Industrial research has produced 3D printers that can print multiple tones and colours; however, these can be expensive [5,6]. In terms of application, not only do the difficulties associated with general colour reproduction have to be dealt with, but in addition to this, colour deposited in 3D has its own rules and perceptual oddities. 2.5D printing, an variation of 3D printing in which two dimensions are much larger than the third, is giving a more manageable access to colour in 3D.

An extremely early 2.5D printing method is the Woodburytype, in which a photographic image is generated by a relief of pigmented gelatin. A Woodburytype image has continuous tone and so varies smoothly across the image. It does not rely on the half-tone dot pattern or structure, and instead uses differing print heights to generate grayscale and colour. The Woodburytype printing process was invented by Walter B. Woodbury and patented in the late 19th century [7,8,9,10] as a photomechanical process. A comprehensive overview of the history and process can be found in [7]. Traditionally, these prints were monochromatic and used a carbon black pigment; however, multi-colour prints can be achieved by layering these translucent sheets selectively to create a stratified optical film. Gelatin, in general, has become an increasingly important material in print applications with the advent of bioprinting [11,12,13,14,15], and so we seek methods that allow for further understanding in the manipulation and deposition of the material.

A Woodburytype print is produced through a four-step process [7,8,9]:A relief printing plate is constructed, with the aid of a CNC mill or photopolymer plate. The depth of the relief dictates the lightness of the print at each point. The deeper it is, the darker the print will be.A gelatin-based medium is mixed and pigmented to create the ink.The printing plate is filled with the gelatin ink, compressed against the substrate and left to set.Once set, the substrate is pulled from the plate and the print is left to dry in ambient conditions.

Figure 1 shows a pictorial version of the general process. After the ink has dried, we can repeat this process to overlay another layer of ink to the print, with either the same or differing pigment content, increasing both the variation in print heights and the complexity of the colour gamut.

In terms of optics, the tone of the print is achieved by variations of the print height, and therefore of the probability that photons are absorbed or scattered as they pass through the printed layers. The higher the attenuation coefficient and the longer the light path through the medium, the darker the tone. Due to the nature of the Woodburytype process, we can only print in layers of a single colour, in a specific order. The combination of the differing layers will give a range of colours, the range of which is determined by the chosen component colours, the printing order and the largest/smallest print height per layer. The work in [9] contains a description of the single layer Woodburytype and an optical model that defines it. We expand this approach to a multi-layer and multi-pigment set of Woodburytype prints and discuss how full colour Woodburytypes can be produced from a reference image.

## 2. Materials

### 2.1. Ink Formulation

The gelatin inks used here consist of 82.5 wt% of deionised water and a combination of gelatin and pigment that sums to 17.5 wt%. Pigment loads are typically very low, due to the relatively large print heights involved, and range from 0.001 to 0.05 wt%. The gelatin used was Rousselot 250 LB 8 (Rousselot, used as received) with a bloom of 250. The amount of the gelatin/pigment mix used is kept at a constant percentage of the total weight as this has a direct impact on the final print height of the dried Woodburytype.

In order to make the ink, half the DI water is added to the gelatin and the mixture left to swell. The pigment is dispersed in the rest of the water and added to the water/gelatin mixture. The final mixture is melted in a low temperature oven at 50 °C and then placed in a fridge at 6 °C for 12 h, before being reheated to 45 °C to ensure proper flow for printing purposes. Gelatin is used here primarily because it forms the historical basis of the ink recipe. It was, and still is, valued for being cheap, abundant and optically very transparent and has an extensive history of use in the photographic and print techniques of the same era [7]. It additionally allows for their prints to hold the overall shape of the relief plate during the drying process. Typically these inks are stored at low temperatures and in a gelled state to dissuade aggregation of pigments and long-stage ageing of the gelatin, which can cause variations in the chemical state of the gelatin [16,17,18].

Three separate ink mixtures are produced—cyan, magenta and yellow. These use Hostaperm Blue BT-617-D (pigment blue 15:4, based on phthalocyanine), Ink Jet Magenta E02 (pigment red 122, based on quinacridone) and Ink Jet Yellow 5GX-W (pigment yellow 74, based on monoazo yellow), respectively. All pigments were obtained from Clariant and used as received. Particle sizes in aqueous dispersions, based on readings from a Zetasizer Nano S, are recorded as 136.9 ± 1.21 nm, 155.0 ± 2.44 nm and 171.0 ± 1.5 nm, respectively. During the ink formulation, Solsperse 27000 is added to improve overall stability and pigment dispersion and Siltech C-22 is added as a mar resist. We can vary the pigment as the Woodburytype places no real restrictions on pigment type; however, it should be taken into consideration that the following model lays out a test case and only holds for these particular chosen pigments and will need to be reformulated if the ink recipes are modified.

### 2.2. Ink Characterisation

We use an Agilent 8453 UV–visible spectrophotometer to obtain transmission and absorbance data of the pigments, in aqueous solution, in the visible range (350–750 nm). This data is then averaged over 5 nm intervals in order to smooth out local variations or noise due to defects. In order to obtain roughly similar absorbance peak intensities, a ratio of pigment concentrations of 0.7:1:1 of the C:M:Y components is required and displays data similar to the ratios seen in Figure 2a,b. As in standard inkjet CMYK colour models, the presence of a K (black) component is required due to the non-perfect combination and overlap of the CMY spectra. The transmission of such a mixture can be seen in Figure 2b which, while fairly flat in comparison the component inks and therefore close to a black ink, is visually more akin to a dark green/brown.

Figure 3a shows an example of the step-wedge Woodburytype prints used to quantify the variation in tone with print height. This consists of a “five-step wedge”, which is devised as adjacent steps, ranging in print height from 0.2 to 1 mm. The final dried print heights arising from this depend on the amount of gelatin used in the formulation, however the increasing print height always results in an increasing perceived darkness. These step wedge prints provide the range of transmission data, seen in Figure 3b–d, used to characterise the print and provide the basis of the colour tracking model.

## 3. Results and Discussion

### 3.1. Multi-Layer Printing Process

In order to print multiple layers of translucent gelatin, there are two possible approaches:Construct plates that colour separate the image and take into account the uneven print surface caused by prior printed layers.“Neutralise” the print topography by adding layers of un-pigmented gelatin to each print layer and ensuring we always print on a flat surface.

We primarily work with the first suggestion as layers of unpigmented gelatin, while very transparent [19], it does cause a non-negligible scattering and absorption that increases toward low wavelengths, and would therefore need to be taken into account when attempting to recreate specific colours. On the other hand, constructing plates that take into account the print layer(s) below, mean that this print order is locked and cannot be altered without constructing new plates. Figure 4 displays an example of the full colour Woodburytype using the adjusted plate method, in comparison to a traditional CMY lithographic print of the same image.

As another test print plate, we expand the step-wedge print to ten equal 0.1 mm steps, ranging from 0.1 mm to 1 mm. These steps are linearly spaced, as opposed to exponentially spaced to counter the decay of light propagation through a material, to ensure that calculation of the final dried print heights is simpler. A second-layer step-wedge plate, consisting of ten steps from 0.1 +Δ mm to 1 +Δ mm, where Δ corresponds to dried print heights left behind by the first print. In [8], we produced an empirical relationship between the step depth and the print height for this specific ink formulation as being
(1)Δ=(0.0136×D)+(0.0230),
where Δ is given in millimetres and *D* is the depth of the step in millimetres. When printed on a rough substrate such as paper, we see a variation of ±8% across a printed “flat” layer. A third layer plate can also be made, ranging from 0.1 +2Δ(D=0.1) mm to 1 +2Δ(D=1) mm.

Figure 5a shows how the second layer print, when designed to accommodate the already dried print layer beneath, allows for this. It was initially thought that the application of the second layer of aqueous gelatin solution to the first dried print may rehydrate the layer and cause deformations in the print height; however, Figure 5a shows that this effect is minimal. The two print layers are clearly defined and separated and display similar print heights of 163 μm and 168 μm. These are within 7% of the value given in Equation (1) and therefore within the standard variation we detect across an intended flat plane of the print [9]. One possible reason for this variation is the imperfect surface of the paper substrate, seen in Figure 5b. These surface variations have characteristic x–y length scales between 20 and 80 μm.

To further facilitate accurate image reproduction, that attempts to fully reconstruct the colours of an input image, we individually scale the component images against the characteristic exponential decay seen in the following Kubelka–Munk model. The method for this is discussed in Appendix A.

### 3.2. Colourimetric Measurements & Optical Models

#### 3.2.1. Kubelka–Munk Theory

Kubelka–Munk theory is an optical model typically used to describe the optical effects of films of ink or pigments. Optically these are described as turbid media, in which the effects of individual photons can be incredibly complex to model and so instead the film is characterised with an absorption *K* and scattering coefficient *S* per unit length [20,21,22]. From these, information such as the colour and reflection/transmission of the film can be inferred and varied with both pigment concentration and film thickness. It is assumed that these coefficients are uniform throughout the thickness of the film and that there are no prominent boundary effects, such that the plane has an effectively infinite width and length.

Kubelka–Munk (KM) is a top-down approach to understanding the optical properties of thin films of paints and pigment. The individual properties of the pigment particles are replaced by an effective descriptor that averages over multiple particles. Attempts to link this overall descriptor with individual particle properties are highly dependent on the exact situation being modelled and, in particular, the ratio between *K* and *S* [23,24,25,26,27]. Extensions of the model exist to give it further and more complex utilisation [28,29,30]; however, we utilise it primarily for the simplicity of the model and the explicit dependence of the optical properties on film thickness, which is the direct cause of variation in tone of the Woodburytype.

The reflectance R(λ) of a film on a substrate can be predicted via the KM coefficients as [25,31,32,33]
(2)$$R=\frac{1-R_{g}\left(a\;-b\coth\left(bSd\right)\right)}{a-R_{g}+b\coth\left(bSd\right)},$$
where
(3)$$a\;=\frac{S+K}{S},\;b=\sqrt{a^{2}-1}.$$

These equations dictate how the reflectance changes with the substrate reflectance $R_{g}$ (that ranges between 0 and 1 for a transparent and specular reflector, respectively) and film thickness, *d*.

An alternate set-up, suggesting a freestanding film, has reflectance and transmission values of
(4)$$R=\frac{\left(1-\beta^{2}\right)\sinh\left(\kappa d\right)}{\left(1+\beta^{2}\right)\sinh\left(\kappa d\right)+2\beta \cosh\left(\kappa d\right)},$$
and
(5)$$T=\frac{2\beta }{\left(1+\beta^{2}\right)\sinh\left(\kappa d\right)+2\beta \cosh\left(\kappa d\right)},$$
where
(6)$$\beta =\sqrt{\frac{K}{K+2S}},\;\kappa =\sqrt{K+2S}.$$

These spectral responses, denoted as ϕ(λ), can be used to calculate the tristimulus values *X*, *Y* and *Z* and their corresponding chromaticity *x* and *y*. The tristimulus values represent three imaginary component colours that can be used to construct any real colour and define the chromaticity, the colour as perceived by the human eye. They are given by [34]
(7)$$X=100\sum_{\lambda }\frac{\phi \left(\lambda \right)I\left(\lambda \right)\bar{x}\left(\lambda \right)\Delta \lambda }{I\left(\lambda \right)\bar{y}\left(\lambda \right)\Delta \lambda },$$
(8)$$Y=100\sum_{\lambda }\frac{\phi \left(\lambda \right)I\left(\lambda \right)\bar{y}\left(\lambda \right)\Delta \lambda }{I\left(\lambda \right)\bar{y}\left(\lambda \right)\Delta \lambda },$$
(9)$$Z=100\sum_{\lambda }\frac{\phi \left(\lambda \right)I\left(\lambda \right)\bar{z}\left(\lambda \right)\Delta \lambda }{I\left(\lambda \right)\bar{y}\left(\lambda \right)\Delta \lambda },$$
where I(λ) is the distribution of the illuminant (we use a D65 illuminant, given as the standard illumination conditions of many outside open-air portions of the world) and $\bar{x}$, $\bar{y}$ and $\bar{z}$ are the colour matching functions, used in the CIE XYZ colour space and seen in Figure 6. As previously mentioned, these can be related directly to the chromaticity coordinates via
(10)$$x=\frac{X}{X+Y+Z},\;y=\frac{Y}{X+Y+Z}.$$

Another alternative representation of the chromaticity, known as CIE $L^{*}a^{*}b^{*}$, is also useful. This changes the colour space into one that is equally spatially distributed, such that a difference of 1 can be considered a change in the colour that is perceptually distinguishable by the human eye [35]. In this, $L^{*}$ represents the luminance or “lightness” of the colour and $a^{*}$ and $b^{*}$ are a description of the colour. Positive $a^{*}$ indicates red components, while negative indicates green and positive $b^{*}$ suggests increasingly yellow components, whereas negative represents blue. These are derived from the CIE XYZ space as
(11)$$L^{*}=116f\left(\frac{Y}{Y_{n}}\right)-16,$$
(12)$$a^{*}=500\left[f\left(\frac{X}{X_{n}}\right)-f\left(\frac{Y}{Y_{n}}\right)\right],$$
(13)$$b^{*}=200\left[f\left(\frac{Y}{Y_{n}}\right)-f\left(\frac{Z}{Z_{n}}\right)\right],$$
where $f\left(t\right)=t^{1/3}$ if $t>\delta^{3}$ or $f\left(t\right)=t/3\delta^{3}+4/29$ if $t<\delta^{3}$. $X_{n}=95.049$, $Y_{n}=100$ and $Z_{n}=108.891$ for the standard illuminant D65. We use a Konica-Minolta FD-7 handheld spectrodensiometer to obtain the CIE data; this limits our range of available wavelengths from 380 to 730 nm.

#### 3.2.2. Colour Tracking with Print Height

The chromaticity values or CIE coordinates can be plotted on the CIE 1931 color space chromaticity diagram to provide a general overview of the varying colorimetric properties [37]. The values for step-wedge prints of cyan, magenta and yellow can be seen in Figure 7, for a weak ink set of 0.00108 wt%, 0.00176 wt% and 0.00176 wt%, respectively. A weak ink set is used as the differences in the chromaticity values between adjacent steps of the wedge are relatively more pronounced for weaker inks than for stronger ones and we do not risk entering the regime of the “infinitely thick” film, whereby all colour is produced by reflections at the surface and the region immediately beneath it. The colour of these infinitely thick films is unchanged with increasing print height and therefore of little use to our model. Alongside this, a dilution of the pigmentation in the ink in general is preferable for multi-layer prints in comparison to their monochromatic counterparts. On the other hand, however, lower concentrations of pigment enhance the scattering properties provided by the gelatin binder [8,9], which we typically seek to suppress as it makes the colour tracking more difficult. Therefore, in practice, a balance will have to be struck between these two effects in deciding the optimal pigment load.

Figure 7 shows how these *x* and *y* values vary with increasing print height, with the starting and ending steps of 0.1 mm and 1 mm being highlighted with crosses. The general trend is the shallowest steps tend toward the white point, while the largest move out toward the outer edges of the chromaticity diagram, that represents pure single-wavelength versions of these colours. As this colour space is not uniform in distribution, the length of lines being unequal does not mean that the intensity of each colour is unequal. Moreover, note that we are concerned purely with the colour components *x* and *y* here; the lightness component *Y* is usually included to give a fuller description of the colour.

Figure 7b highlights further the chromaticity values of the double-layer prints, comprised of printing a single colour step-wedge and then printing over the second layer step-wedge of a differing colour. These are shown by the second-order colour lines of red, green and blue. Typical colour mixing rules state that for any two points on the chromaticity diagram, the colours obtained by mixing them lie somewhere on the line connecting them. However, here, these double-layer prints present purer and more intense colours as they use double the print heights of the single layer CMY prints.

Choosing to make multi-layered prints means we must also make a decision on the order of the printing and, particularly, discern whether this has any direct effect on the final colour produced. The CIE xyY colour space is useful for tracking how colour mixing occurs; however, it is not perceptually uniform [38,39] and so we instead use the CIE $L^{*}a^{*}b^{*}$ space to track colour differences. Figure 7b shows the double-layer prints for one order with the solid line and the opposite order (e.g., CM→MC) as a dashed line. In each case, a difference in the printing order produces colours that are perceptively different at each step, such that
(14)$$\Delta E^{*}=\sqrt{{\left(\Delta L^{*}\right)}^{2}+{\left(\Delta a^{*}\right)}^{2}+{\left(\Delta b^{*}\right)}^{2}}>1.$$

This perceptive difference increases with step height and is pushed toward the colour that is printed last—the uppermost layer. Figure 8 shows this effect, by tracking the $\Delta E^{*}$ value with print height. This suggests that there is some complex internal scattering effect, caused by either the pigment, gelatin, substrate or some combination of all of them, directly affecting the colour output.

#### 3.2.3. Empirical Model

The Woodburytype allows for the fast layering of multiple coloured films, and therefore a direct and quick empirical model that links the colour of the print to the print heights of the layers below is desirable. As such, we test the boundaries of a simplistic model that aims to fit lines of best fit to the single colour cyan, magenta and yellow prints and then further utilise those in order to predict the double-layer prints. This approach works as, traditionally, the combination of any two additive colours within the xyY colour space can simply be found on the line joining those two colours [40]. We have determined in Figure 5 that each step in the double-layer print is simply double the print height of a single layer print, and so we should be able to extrapolate these results to obtain the combination colours in a similar manner, scaling by the doubled print height.

Fitting to the single layer prints in Figure 7 we obtain
(15)$$\left[\begin{array}{c} x_{C,M,Y}\left(D\right)\\ y_{C,M,Y}\left(D\right) \end{array}\right]\approx \left[\begin{array}{c} m_{C,M,Y}^{x}D+c_{C,M,Y}^{x}\\ m_{C,M,Y}^{y}D+c_{C,M,Y}^{y} \end{array}\right],$$

It is worth noting that the values of $m_{C,M,Y}^{x,y}$ and $c_{C,M,Y}^{x,y}$ apply purely at this concentration and so new relations will have to be derived for each new ink set. Simply by treating the double-layer print as an average of the two component prints, with a doubled print height, we can obtain a fitting that predicts these double layer colours for each step. For example, the xy values of the red prints can be estimated from the data of the magenta and yellow prints as
(16)$$\left[\begin{array}{c} x_{R}\\ y_{R} \end{array}\right]\approx \frac{1}{2}\left[\begin{array}{c} x_{M}\left(2D\right)\\ y_{M}\left(2D\right) \end{array}\right]+\frac{1}{2}\left[\begin{array}{c} x_{Y}\left(2D\right)\\ y_{Y}\left(2D\right) \end{array}\right].$$

The results of this simple fitting can be seen in Figure 9. Prints in which each component layer has a different height can be modelled simply by changing the scaling given by the 1/2 term that determines the ratio between the two. We should expect this model to further deviate from the experimental results with increasing concentration and print height as it ignores many complexities of the problem, such as the increasing chance of defects and the larger difference between printing orders. Moreover, again, this ignores the lightness component *Y* required for the full description of a colour in the CIE xyY model. However, this does allow for a very quick and rough solution to the colour of a multi-layer Woodburytype print.

#### 3.2.4. K(λ) and S(λ) Determination

Using the general interactions we have learned from the empirical model, we instead turn to a more complete KM model to describe the colour of the multi-layer Woodburytype. We first print the step-wedge prints onto a transparent glass slides. This allows for easy measurement of the transmission of the freestanding film, without any interference from a substrate layer. These are again tracked to the layer thickness *d* by the equation
(17)$$T=\frac{2\beta }{\left(1+\beta^{2}\right)\sinh\left(\kappa d\right)+2\beta \cosh\left(\kappa d\right)}.$$

In each case, a similar approach to the black pigment method is used [9], whereby a vast matrix of *K* and *S* is searched for the closest fit to the spectra, individually per wavelength. Each value is then scaled by the volume fraction *f*, in an attempt to make these fits more general across multiple pigment concentrations [41]. We use minimisation of the mean square error (MSE) to determine the closest fitting for each of the three tristimulus values X,Y and *Z*. For example,
(18)$$MSE_{X}=\frac{1}{n}\sum_{1}^{n}{\left(X-\bar{X}\right)}^{2}.$$
where n=5, corresponding to the five steps of the wedge print. We essentially seek to find the values of K/f and S/f that minimise $\sum MSE_{X,Y,Z}$. Search time, in general, is reduced by utilising this large database at 50 nm intervals and then using those solutions to inform nearby values of a smaller search database, centred around those values. This ensures the chosen K(λ) and S(λ) vary more smoothly. These chosen values are fed back into Equation (5) to provide a comparison of a calculated spectra against the measured spectra.

As the Kubelka–Munk method relies primarily on a generalised combination of absorption and scattering terms, it is important to understand the system being modelled, as varying combinations of K/f and S/f can lead to the same result. This is shown for transmission *T* in Figure 10. The pigments used here are of the order of 150 nm in size, small enough to favour absorption over scattering. These are chosen in an attempt to suppress the more complex optical properties that would be provided by highly scattering particles. However, as all of our pigments have regions of the visible spectrum that allow for large transmittance, the optical properties of the gelatin binder become more apparent and this results in the introduction of more scattering to the system—as was found in low pigment load samples in [9].

The general trend of these fittings show large absorption values that vary with wavelength, in accordance with the transmittance values. Regions of transmission that vary less with print height are more influenced by the scattering term. This is the previously discussed, optically “transparent”, region where the light is unaffected by the pigment and instead interacts somewhat with the gelatin in the ink. The authors of [23] discuss *K* and *S* values for quinacridone particles (the basis of our magenta pigment), but for much larger particle sizes of ∼500 nm and aggregations. In that case, scattering dominates over the absorption due to size effects; however, it does help pin the absorption and scattering peaks in roughly the same wavelength regions as our fit.

The regions of greatest difference between the fit and the data exist at the largest transmission values, as this corresponds again with more prevalent scattering regions in which the colour and lightness values become harder to predict [23]. The small differences between steps at certain wavelengths mean they are more sensitive to general measurement error, in comparison to their overall magnitude. This all results in multiple combinations of K/f and S/f that can produce similar values of optical extinction and are very easily affected by small variations in the data.

#### 3.2.5. Colour Prediction

Using the *K* and *S* coefficient relationships, determined per wavelength in Figure 11, we can begin to predict the variation in the colour of the print with changing print height. In our model, we make the assumption that due to use of the same binder material and same drying conditions, the refractive index change between multiple layers should be negligible. In addition to this, we assume the light enters the other layers in the same manner as when the light enters the top layer from the air. This is reductive, but the high absorption properties of the pigment suppresses the scattering for the majority of the photons entering the print, and therefore we can assume the colour should be driven primarily by the absorption component and therefore the pigment itself.

Figure 12 shows the results for the three component inks and Table 1 presents the experimental data. Variation between the data and the model can be attributed to the assumptions listed above and the two main sources of measurement error; local variation due to the substrate topography (the spectrophotometer set-up we use has an aperture that is 3.5 mm in diameter) and the probability of non-visible defects occurring in the print increasing with print height. Due to the additive nature of the KM model, the accumulation of error should mean that the model will naturally drift further from the true values with increasing print height. Further optimisation of the model could be achieved by varying the pigment concentration and repeating this process to obtain a more general model, as seen for the method of characterising the black pigment [9], or simply including a broader range of transmission against print height data in the original search for the *K* and *S* values.

As seen in the empirical model, the print order can have a direct effect on the final colour of the print. In order to test whether this was purely a consequence of scattering at the reflection of the substrate or an inherent effect in the colour ordering, we printed the step-wedge prints onto a transparent substrate and measured the transmission at multiple print heights. Figure 13 shows the results for one such colour combination. We find in this situation that the colour closest to the surface has the more pronounced effect on the measured colour. From this we can determine there are indeed multiple competing effects emerging here, the perceived colour shift due to the print order and the perceived colour shift due to scattering processes induced by the reflectance from the substrate. As these are seemingly both at play and the second will vary dramatically on the type of substrate used, we can only attempt to suppress this effect by using large concentrations of highly absorbent pigment.

### 3.3. Colour Space

Figure 14 shows the predicted colour space by the model for combinations of the three inks, in the range of 0 to 160 μm per layer (corresponding to a plate depth of 0–1 mm). Also shown, in filled red circles, is the *x* and *y* coordinates of the single and double step-wedge prints for the largest possible print height. Comparing these two shows that the full Kubelka–Munk model provides a reasonable representation of the colour space available to the full colour Woodburytype process, bounding the single and double-layer prints fairly closely. The third-order prints are not documented as a large print height combination of all three prints produces a dark brown/black shade (one sample displayed $L^{*}=10.08,a^{*}=-15.07,b=12.75$), a colour that does not have much mathematical value in the CIE xyY colour model, but may account for some of the theoretical values lying outside the suggested experimental colour space. When recreating a reference image, colours that lie outside this gamut instead will be approximated as the extremal value of it, as is typical when transferring images between nonequivalent colour gamuts.

## 4. Conclusions

In order to produce a high-quality, polychromatic, 2.5D print in the form of the Woodburytype, we tracked the optical and colourimetric response of multiple layers of translucent gelatin, coloured with a variety of pigments. We used the basis of a CMY ink set to produce a large colour gamut, in a similar way to traditional inkjet prints. The model that informs this is built from the CIE $L^{*}a^{*}b^{*}$ response of the prints, which varies with both print height and concentration. We discussed how a very simple empirical model, based on the chromaticity values in the CIE xyY colour space, can be used to predict the colour at low pigment concentrations and then expanded to a more comprehensive Kubelka–Munk model, to define the colour space limitations of the generalised full-colour Woodburytype.

## Figures and Tables

**Figure 1 molecules-25-02468-f001:**
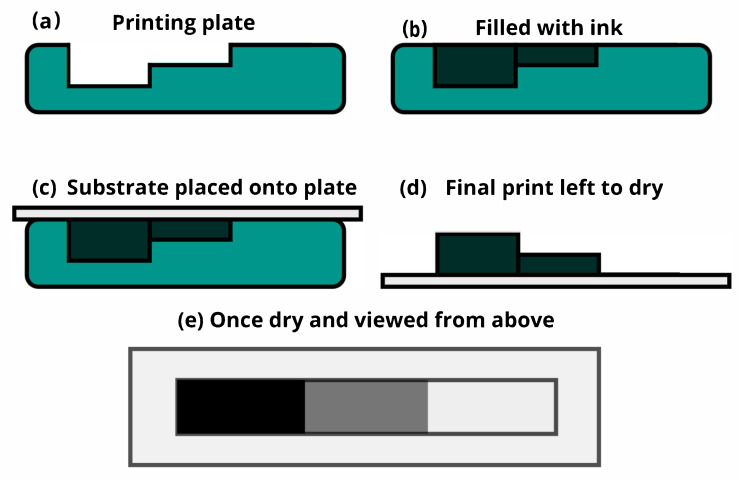
A pictorial version of the general Woodburytype process. (**a**) A relief printing plate is produced, with depths to indicate changes in tone. (**b**) The relief plate is filled with the gelatin ink—recipe described in the text. (**c**) The plate and ink are pressed against the desired substrate and allowed to set. (**d**) Once set, it is pulled from the plate and allowed to dry under ambient conditions. (**e**) A example of how the tone arises from the varying heights of the films of translucent ink mixture is shown.

**Figure 2 molecules-25-02468-f002:**
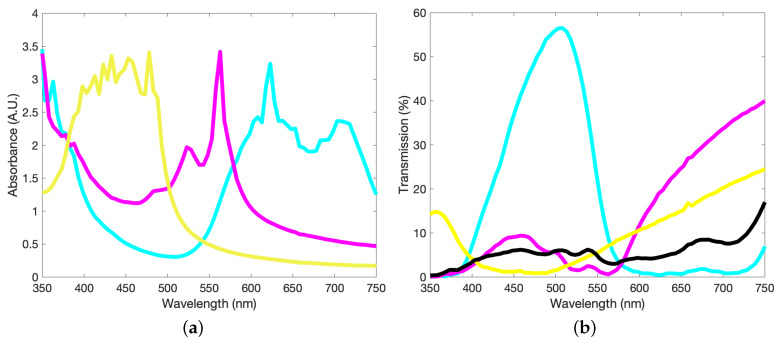
(**a**) Absorbance values for the cyan, magenta and yellow pigments in aqueous solution. We average over a 5 nm interval to minimise the noise of the output. No gelatin is present here; the inks are derived from a diluted version of the recipe detailed in the text. (**b**) Corresponding transmission values of the three pigments. Also shown in black is the transmission of an equal mixture of 1/3:1/3:1/3 of the three pigments, included to show the “near-black” alternative provided a direct combination of the three colours.

**Figure 3 molecules-25-02468-f003:**
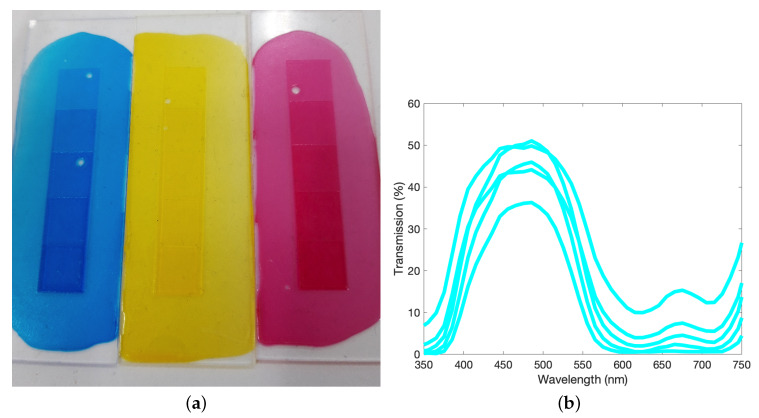
(**a**) Examples of five-step prints constructed from the CMY inks, varying in print height from 0.2 mm to 1 mm and printed onto a transparent glass slide. (**b**–**d**) Transmission values for the three pigments printed onto a transparent substrate, averaged over 5 nm intervals. These inks contain a pigment load of 0.037 wt%, 0.035 wt% and 0.035 wt%, respectively. The largest magnitude transmittance curve represents the smallest print height and the smallest magnitude transmittance curve represents the largest print height. Some overlaps in the orders occur due to defects arising in the print process.

**Figure 4 molecules-25-02468-f004:**
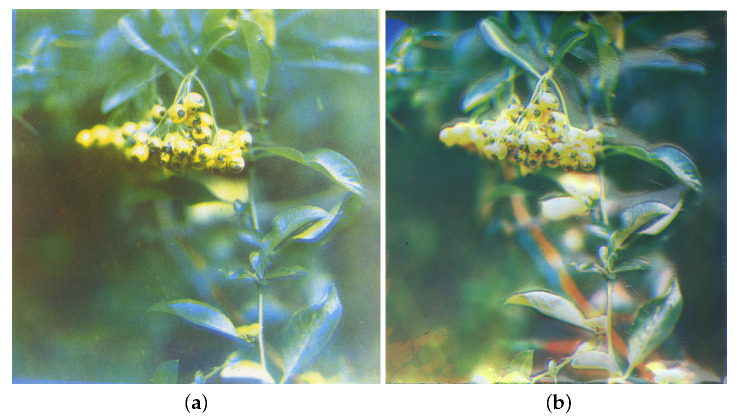
Scans of a test image, printed using both (**a**) traditional lithography techniques and (**b**) the Woodburytype process. The Woodburytype image is built using the method defined in the text, from a three-component CMY print process. It displays a similar level of clarity, definition and colour range to the well-documented lithography method, despite being a two-dimensional representation of an image with physical variation in depth.

**Figure 5 molecules-25-02468-f005:**
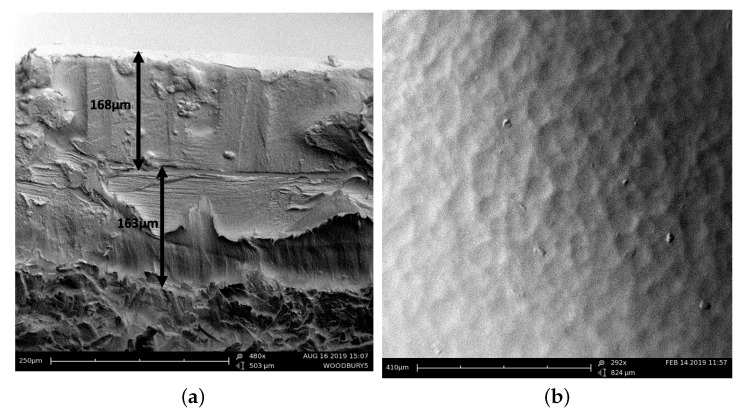
(**a**) SEM image of the cross section in a double-layer red print, ordered as MY, showing two well-defined print layers are of similar height. This is due to the built-in accommodation provided by the second-layer printing plate, as described in the text. (**b**) Image of the paper surface, showing the uneven substrate. The feature size, averaged over fifteen measures, is 61 μm in diameter. This could be one of the immediate causes for variation in print height across a flat plane of print.

**Figure 6 molecules-25-02468-f006:**
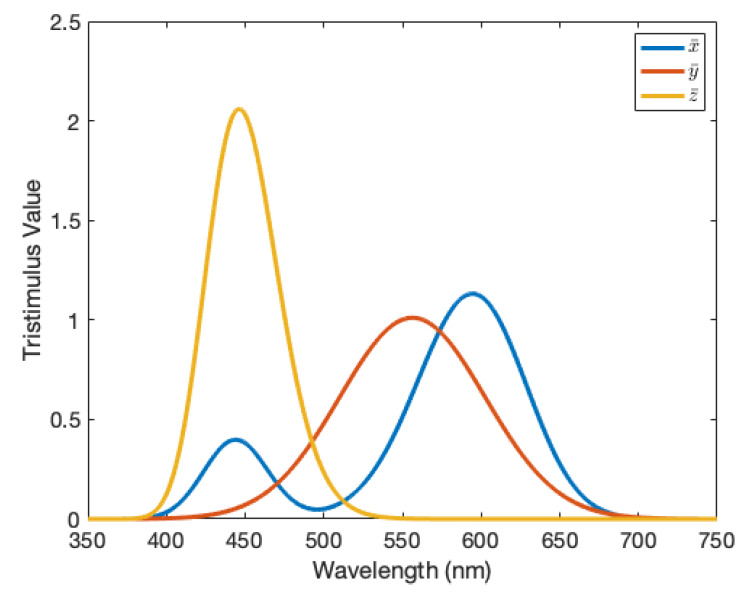
The colour matching functions $\bar{x}\left(\lambda \right)$, $\bar{y}\left(\lambda \right)$ and $\bar{z}\left(\lambda \right)$ of the standard illuminant D65 for a standard colourimetric observer [36].

**Figure 7 molecules-25-02468-f007:**
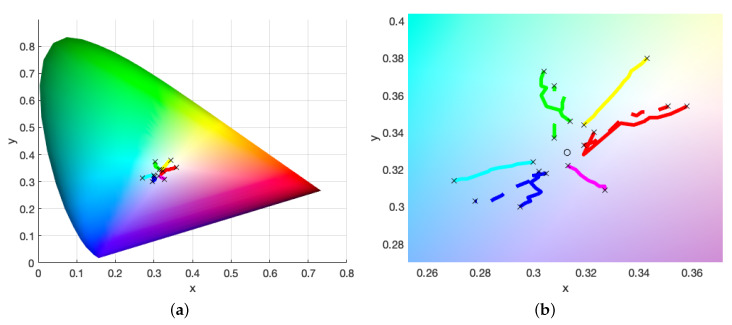
(**a**) CIE xyY values for a set of extremely weak Woodburytype inks, showing the three single layer prints possible with CMY inks. The inks chosen are weak as the multi-layering process here effectively doubles the print height range with each layer that is added. The region of “infinitely thick” films whereby all colour is produced by reflectance at the surface, is easily reached by the Woodburytype process due to the increasing print height. The intended print colour is indicated by the line colour, for instance, cyan for the cyan ink and so on, and the minimum and maximum steps are highlighted with black crosses to stand out from the background further. The double-layer prints, resulting in red, green and blue, are also shown. (**b**) The same data, but zoomed in further for clarity. The white point, x=0.31271,y=0.32902, is highlighted with a circular point, alongside the equivalent double-layer prints that are switched in print order (i.e., CM→MC), represented by the dashed lines of the same colour.

**Figure 8 molecules-25-02468-f008:**
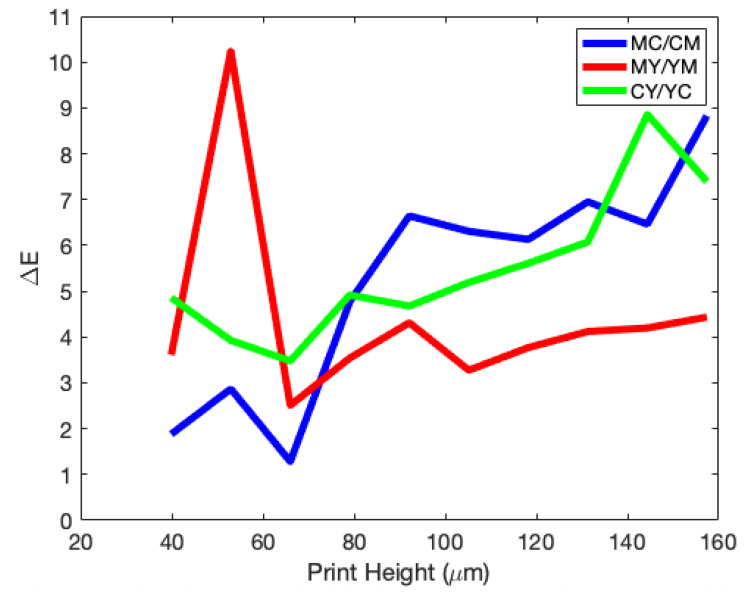
$\Delta E^{*}$ values, the degree of difference between two prints, at each step height for the three second-order colours (RGB), when swapping the printing order (e.g., CM⟷MC). If ΔE is larger than one, then the two prints are perceptively different.

**Figure 9 molecules-25-02468-f009:**
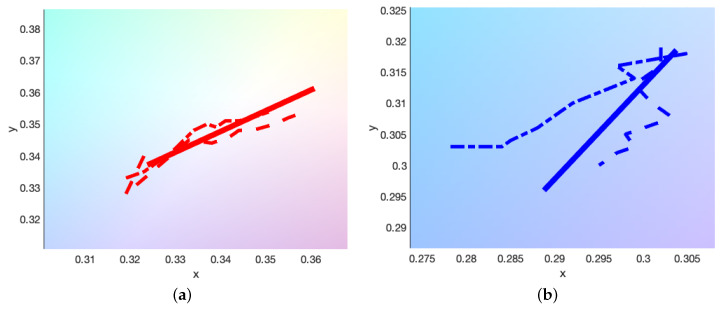
Empirical fitting of double-layer print xy values, based on a simple combination of the constituent colour xy values, for red (**a**) and blue (**b**). The solid line represents the empirical fitting and the two dashed lines represent the data provided by both of the colour combinations (e.g., CM and MC).

**Figure 10 molecules-25-02468-f010:**
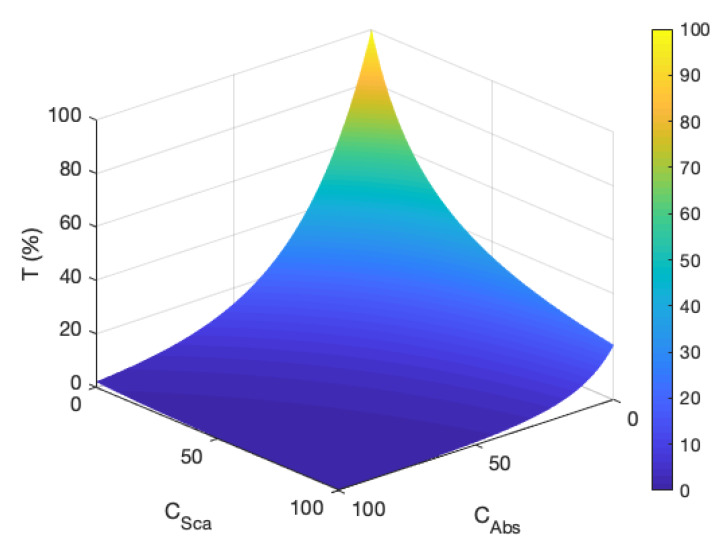
Transmission values with changing $C_{Sca}$ and $C_{Abs}$ (μm). As the extinction coefficient (the sum of the two) increases, the transmission decreases, however differing combinations of $C_{Sca}$ and $C_{Abs}$ can produce similar values. Therefore, we require more knowledge of the underlying physics of the problem in order to properly describe the transmission through the gel films.

**Figure 11 molecules-25-02468-f011:**
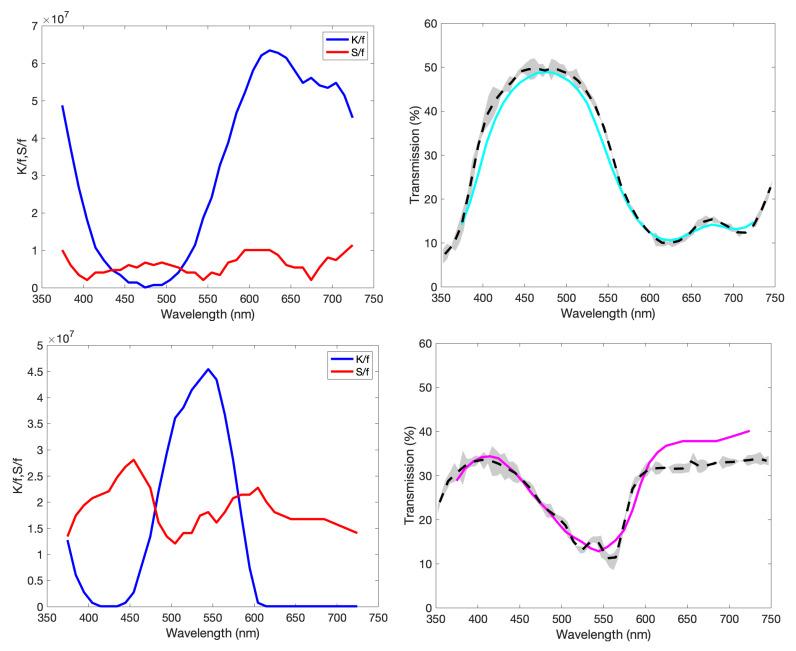
K/f and S/f values, varying with wavelength, chosen to match the transmittance of each spectra. Alongside these are the measured (dashed) and predicted (solid) transmittance for cyan (top), magenta (middle) and yellow (bottom). Also shown is the standard deviation of the measured transmittance. Here, we have scaled by the volume fraction to attempt to make the fit more general.

**Figure 12 molecules-25-02468-f012:**
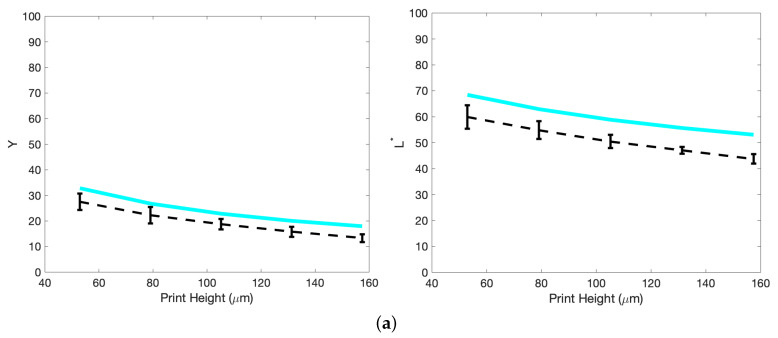
Results and corresponding predicted fitting for the lightness components, *Y* and $L^{*}$, of the step-wedge prints. These are listed individually as (**a**) cyan, (**b**) magenta and (**c**) yellow. The dashed black line represents the data taken from the prints averaged over three measurements, while the solid line represents the predicted curve, with absorption *K* and scattering *S* coefficients chosen for the greatest fit.

**Figure 13 molecules-25-02468-f013:**
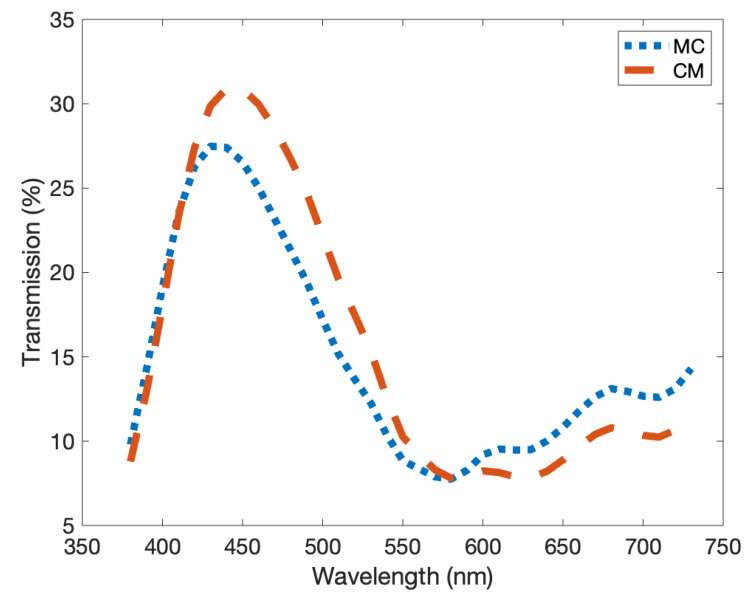
The predicted transmission of two second-order prints of cyan and magenta and the reverse order. The peak value moves from 30% at 440 nm to 27.5% at 430 nm. Note how the colour printed last influences the character of the transmission more strongly.

**Figure 14 molecules-25-02468-f014:**
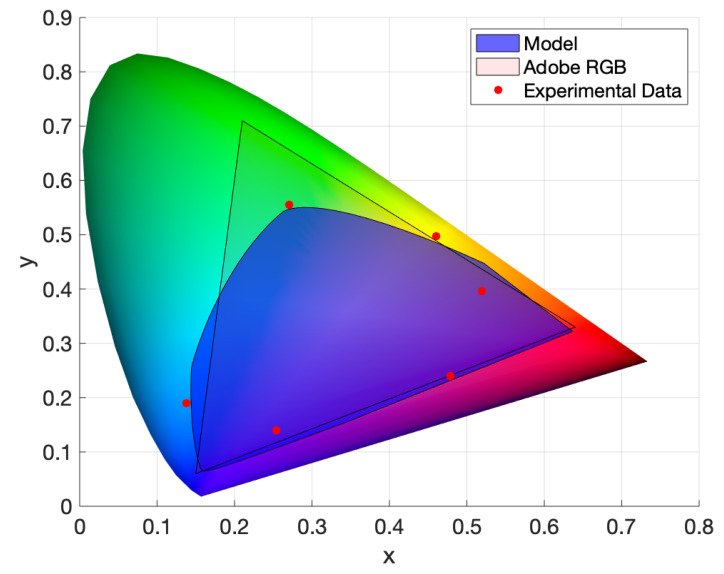
The predicted colour space from the model fitting. This shows the bounds on the *x* and *y* values and the colours we can produce, using the ink formulation in Figure 11 and restricting each component layer to a plate range of 0 to 1 mm. Note that these inks hold a much larger colour gamut than those in Figure 7. The Adobe RGB colour space that represents the colour space possible for a generic CMYK inkjet print is given as a comparison. The solid red markers display the *x* and *y* values obtained from experimental data for single and double prints, displaying the highest print height and therefore “strongest” version of that colour.

**Table 1 molecules-25-02468-t001:** CIE $L^{*}a^{*}b^{*}$ data for the three single layer prints, averaged over three measurements.

	Cyan			Magenta			Yellow		
**Print Height (mm)**	$L^{*}$	$a^{*}$	$b^{*}$	$L^{*}$	$a^{*}$	$b^{*}$	$L^{*}$	$a^{*}$	$b^{*}$
0.049	59.96	−36.20	−38.90	60.46	53.61	−20.40	81.43	−12.30	42.65
0.075	54.83	−34.03	−44.17	52.57	67.05	−20.14	77.92	−15.10	60.11
0.101	50.46	−30.41	−47.43	47.17	71.18	−15.04	75.62	−16.48	74.69
0.127	47.08	−26.08	−49.21	43.89	71.23	−9.48	72.93	−17.02	87.68
0.154	43.79	−22.04	−49.89	41.73	69.50	−3.67	70.58	−16.92	92.77

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
