# Peer review of "The Polychromatic Woodburytype—Colour Tracking in Translucent, Patterned Gelatin/Pigment Films"

_molecules, 2020, doi:10.3390/molecules25112468_

Round 1
Reviewer 1 Report
In the manuscript titled “The PolychromaticWoodburytype - Colour Tracking in Translucent, Patterned Gelatin/Pigment Films”, the authors developed a polychromatic printing strategy using the Woodbury method. To assist the design, they also developed a model to relate the optical response to parameters such as thickness, pigment, etc. In general, this is an interesting topic and their study is reasonably designed. This is publishable with several minor revisions, as listed below:
- It’s highly suggested the author provide an illustration of the Woodbury type process. Although they gave some references for further information of this technique. It’s still useful to include an illustrative figure of the process to make the manuscript self-consistent.
- Line 72: parameters of L*, a*, and b emerge before their definition.
- Many typos. For example, in the first page, there are already several typos.
- Line 16: They are -> There are
- Line 17: a 3D printers -> a 3D printer
- Line 30: to created a …
Author Response
Many thanks to the reviewer for their insightful comments.
Here are our responses to their individual points:
- It’s highly suggested the author provide an illustration of the Woodbury type process.
- This is a great suggestion - we have introduced a new Fig. 1, to give a simple pictorial illustration of the overall Woodburytype process. Along side, some further introductory text to aid it. As stated by the reviewer, this should hopefully allow for a more self-contained understanding of the process.
- Line 72: parameters of L*, a*, and b emerge before their definition.
- We have moved this piece of information further into the text, after the introduction of CIE L*a*b*.
- Many typos.
- We have re-written small amounts of text throughout the manuscript, in the hope of eliminating typos and improving overall readability.
Many thanks,
Damien Leech, Walter Guy, Susanne Klein
Reviewer 2 Report
- Please cite more currently related studies in Molecules journal.
- The Fig. 3 quality need to be improved.
- It is needed to provide more discussions for the gelatin applied roles.
- It is needed to update to new references as the cited references in this article are too old.
Author Response
Many thanks to the reviewer for their insightful comments.
Please find our responses to individual comments below:
- Please cite more currently related studies in Molecules journal.
- Ref. [10, 11, 12] are added to discuss the contribution gelatin has in other current printing methods, with a focus on bioprinting. Ref. [35, 36] have been added as further references for colorimetric definitions.
- The Fig. 3 quality need to be improved.
- The edges of the images have been cleaned up, trimmed and equally scaled. Please note this is now Fig. 4.
- It is needed to provide more discussions for the gelatin applied roles.
- A description of the reasoning behind the use of gelatin has been added in the ink formulation section. It is chosen primarily for being a part of the historical process, but it also contains many material properties that make it useful in this particular process.
- It is needed to update to new references as the cited references in this article are too old.
- More up-to-date references have added, including Ref. [1], [6] and [22]. Some older references have been removed. Where certain references were not possible to remove, such as the original definitions of Kubelka-Munk theory or MacAdam ellipses, newer references have been added alongside to back them up. These include Ref. [21] and Ref. [38].
Many thanks,
Damien Leech, Walter Guy, Susanne Klein
Round 2
Reviewer 2 Report
- Authors have corrected most of the problems that I asked in my previous revision.
- Some experimental quantitative data could be provided if possible and standard deviation of data (even statistical analysis) could be provided.
- Finally, the authors could check English language before publication.
Author Response
Many thanks for the reviewer comments - we have responded to these by:
1) Including standard deviation values in Figure 11 and 12 and including Table 1, that displays the CIE L*a*b* values for a particular set of inks, varying with print height.
2) Small sections of the text have been re-written in an attempt to improve readability.